# 0.5-V Nano-Power Shadow Sinusoidal Oscillator Using Bulk-Driven Multiple-Input Operational Transconductance Amplifier

**DOI:** 10.3390/s23042146

**Published:** 2023-02-14

**Authors:** Fabian Khateb, Montree Kumngern, Tomasz Kulej, Mohammad Yavari

**Affiliations:** 1Department of Microelectronics, Brno University of Technology, Technická 10, 601 90 Brno, Czech Republic; 2Faculty of Biomedical Engineering, Czech Technical University in Prague, Nám. Sítná 3105, 166 36 Kladno, Czech Republic; 3Department of Electrical Engineering, Brno University of Defence, Kounicova 65, 662 10 Brno, Czech Republic; 4Department of Telecommunications Engineering, School of Engineering, King Mongkut’s Institute of Technology Ladkrabang, Bangkok 10520, Thailand; 5Department of Electrical Engineering, Czestochowa University of Technology, 42-201 Czestochowa, Poland; 6Integrated Circuits Design Laboratory, Department of Electrical Engineering, Amirkabir University of Technology (Tehran Polytechnic), Tehran P.O. Box 15875-4413, Iran

**Keywords:** shadow oscillator, shadow filter, analog circuit, operational transconductance amplifier

## Abstract

This paper presents a low-frequency shadow sinusoidal oscillator using a bulk-driven multiple-input operational transconductance amplifier (MI-OTA) with extremely low-voltage supply and nano-power consumption. The proposed oscillator is composed using two-input single-output biquad filter and amplifiers. The condition and the frequency of oscillation of the shadow oscillator can be controlled electronically and independently using amplifiers. The circuit is designed in Cadence program using 0.18 µm CMOS technology from TSMC. The voltage supply is 0.5 V and the power consumption of the oscillator is 54 nW. The total harmonic distortion (THD) of the output signals is around 0.3% for 202 Hz. The simulation results are in accordance with theory.

## 1. Introduction

Shadow filters (or agile filters) are techniques used to enhance the tuning flexibility of a second-order filter by adding external amplifiers in the feedback part [1,2]. The natural frequency and the quality factor of shadow filters can thus be controlled by external amplifiers. There are many shadow filters using various active devices available in literature, such as the current-feedback operational-amplifier (CFOA) in [3], current differencing transconductance amplifier (CDTA) in [4,5], operational floating current conveyor (OFCC) in [6], differential difference current conveyor (DDCC) in [7], voltage differencing differential difference amplifier (VDDDA) in [8], and voltage differencing transconductance amplifiers (VDTAs) in [9,10,11].

Sinusoidal oscillators are widely used in telecommunications, electronics, instrumentation, and control systems—for instance, providing a carrier signal for modulation in telecommunications and control systems [12], or acting as a waveform generator in electronic and instrument systems [13]. Therefore, a number of sinusoidal oscillators have been reported in literature that are based on various active devices such as the commercially-available ICs in [14], second-generation current conveyor (CCII) in [15], CFOA in [16], current differencing buffered amplifier (CDBA) in [17,18], current gain amplifier in [19,20], and voltage differencing inverting buffered amplifier (VDIBA) in [21]. The sinusoidal oscillators that enjoy electronic tuning capability are required because variant generated frequencies can be obtained without changing circuit components, also or the frequency error caused by temperature or process variations can be easily compensated. Usually, sinusoidal oscillators are realized by second-order filters based on two integrators. The internal components used to realize the integrators, such as resistor *R*, transconductance *g_m_*, and capacitor *C*, are used to adjust the frequency of oscillation. Unfortunately, adjusting these components changes the magnitude of transfer function of the integrator and results in changing the amplitude of the output signal.

Recently, a new technique to adjust the oscillation frequency without affecting the amplitude of the output signal have been proposed—the so-called shadow oscillator [22]. It was based on shadow filters [1,2]. This technique uses external amplifiers to control the condition and the frequency of oscillation without changing the internal components of the oscillator, and thus the magnitude of the core network such as integrators is not disturbed. As a result, the performances of shadow oscillators with constant output signal amplitude have been reported in [22,23,24].

Low-frequency oscillators are a part of many biomedical and control instrumentation systems—for examples, see [25,26,27,28,29]. However, these circuits do not employ the shadow oscillator and are unsuitable for the extremely low-voltage and low-power applications seen nowadays. Therefore, the shadow oscillator using multiple-input operational transconductance amplifiers (MI-OTAs) is proposed in this paper. It will be shown that a shadow oscillator can be obtained easily using MI-OTAs. The MI-OTA will be used to realize multiple-input biquad filter and amplifiers to compose the shadow oscillator with minimum components. The condition and the frequency of oscillation of the shadow oscillator can be controlled electronically and independently. It is worth noting here that the MI-OTA has been used for many interesting applications in the literature [30,31,32,33,34,35,36,37] and that the low transconductance offered by the proposed MI-OTA is also desired for biosignals processing in order to achieve a very large time constant of the G_m_–C filter; otherwise, a large chip area will be occupied by the integrated capacitor [38,39,40].

## 2. Proposed Circuit

### 2.1. Proposed Low-Voltage MI-OTA

The electrical symbol of the MI-OTA is shown in Figure 1. In general, the circuit possesses *n* differential inputs, denoted as *V*_+i_ − *V*_−i_, *i* = 1,… *n*. In the particular version discussed in this work, *n* = 2. For this case, the output current of an ideal MI-OTA can be expressed as:(1)$$I_{out}=g_{m}\left(V_{+1}+V_{+2}-V_{-1}-V_{-2}\right)$$

The CMOS structure of the bulk-driven (BD) MI-OTA is shown in Figure 2a. It was first presented with a single current output in [30]. A simplified version of this OTA was also verified experimentally in [31,32]. Its input stage exploits a source degenerative linearization technique, first proposed for a gate-driven (GD) circuit operating in strong inversion; however, in a modified version proposed in [30], the gate-driven transistors were replaced by BD ones, that allows increasing both the linear and common-mode range of the circuit. Like in its GD counterpart, the transistors M_1_ and M_2_ form a differential pair, while transistors M_15_ and M_16_, operating in a triode region, introduce a negative feedback that leads to better linearity of the structure. The multiple inputs of the differential pair (MI-OTA) are realized by replacing the input transistors M_1_ and M_2_ with multiple-input BD devices, as shown in Figure 2b. The multiple inputs of these devices were realized using input capacitive dividers composed of the capacitors *C_B_*. Their values should be considerable larger than the parasitic capacitances of transistors M_1_ and M_2_. These *C_B_* capacitors could be realized on chip in any CMOS technology. In this work, the high reliability Metal-Insulator-Metal capacitors (MIM) offered by TSMC were used.

In order to properly bias the bulk terminals of the input devices for DC, the anti-parallel connections of the minimum-size transistors M_L_, operating in a cut-off region, with *V*_GS_ = 0, were used.

The optimum linearity of the structure is obtained for *m* = (W/L)_15,16_/(W/L)_1,2_ = 0.5 [30]. The rest of the structure can be considered as current mirror OTA, with current mirrors M_3S,D_-M_12S,D_. In order to realize a second current output of the structure, required in the considered application, the composite self-cascode transistors M_8S,D_ and M_11S,D_ were doubled by M_9S,D_ and M_12S,D_, respectively. In a single-output version, the additional transistors should be removed. Due to the fact that the voltage gain of the overall structure was lowered by the application of bulk-driven transistors with lower transconductance, and an input capacitive divider, MOS transistors in current mirrors were replaced by composite self-cascode transistors M_iD_-M_iS_, *i* = 3–14. This allows increasing of the output resistance, and consequently the DC voltage gain, without sacrificing the output voltage swing of the OTA. In addition, in order to further increase the overall voltage gain, a partial positive feedback was applied at the load of the first stage. The partial positive feedback is formed by the cross-coupled transistors M_5S,D_-M_6SD_, which create negative conductances, applied to the drain terminals of the diode-connected self-cascode transistors M_3S,D_ and M_4S,D_, respectively. This lowers the resulting conductances at the drain terminals of M_3S,D_ and M_4S,D_, thus increasing the current gain of the current mirrors M_3S,D_-M_8S,D_ and M_4S,D_-M_8S,D_, and consequently the transconductance and voltage gain of the OTA. Note that in order to maintain the circuit stability in any circumstances, including process, voltage and temperature variations, as well as transistor mismatch, the transconductances of the transistors M_5S,D_-M_6SD_ should always be lower than the transconductances of M_3S,D_ and M_4S,D_. Additionally, since the circuit sensitivity increases as the difference of transconductances *g_m_*_3,4SD_-*g_m_*_5,6SD_ decreases, this difference should not be too small, to maintain both the circuit sensitivity to mismatch, as well as the voltage excursion at the drains of M_3,4SD_, at an acceptable level. Thus, the relationship between transconductances of *g_m_*_5,6*SD*_ and *g_m_*_3,4*SD*_ is a result of a tradeoff between the circuit sensitivity to transistor mismatch, linear range and voltage gain. In the considered design it was assumed *g_m_*_5,6*SD*_ = 0.3*g_m_*_3,4*SD*_, which increases the overall voltage gain by around 3 dB. Assuming unity gain current mirrors, and neglecting second-order effects, the OTA transconductance, determined from one differential input, can be expressed as follows:(2)$$g_{m}=\beta_{i}\eta \cdot \frac{4m}{4m+1}\cdot BF\cdot \frac{I_{set}}{n_{p}U_{T}}$$
where *n_p_* is the subthreshold slope factor for a p-channel MOS, *U_T_* is the thermal potential, *η* = (1 − *n_p_*) = *g_m_*_b_/*g_m_* is the ratio of the bulk to gate transconductances of a p-channel MOS at the operating point, and the attenuation factor *β*, represents the signal attenuation from the input capacitive divider. Assuming that the parasitic capacitances of the MOS transistors are small in comparison with *C_B_*, and for the frequency of the signal the impedances of capacitors *C_B_* are much smaller than the impedances of the biasing transistors M_L_, the coefficient *β* can be expressed as:(3)$$\beta_{i}=\frac{C_{Bi}}{\sum_{i=1}^{2}C_{Bi}}=0.5$$

The coefficient *BF* in (2) represents the transconductance boosting factor coming from the partial positive feedback discussed above, which, neglecting the impact of output resistances of MOS transistors, can be approximated as:(4)$$BF\;\frac{g_{m7,8SD}}{g_{m3,4SD}-g_{m5,6SD}}$$
where *g_miSD_* represents a transconductance of a self-cascode composite transistor M_iS-_M_iD_. Note that in a weak inversion region, the transconductances are proportional to the biasing current flowing through corresponding devices. As can be concluded from (2), the overall transconductance is proportional to the biasing current I_set_, and can easily be regulated using this current.

The low-frequency voltage gain of the OTA can be expressed by:(5)$$A_{VO}≅g_{m}[\left(g_{m9D}r_{ds9D}r_{ds9S}\right)||\left(g_{m12}r_{ds12D}r_{ds12S}\right)]$$

Thanks to all the techniques discussed above, the value of this gain can exceed 30 dB, despite the low voltage supply 0.5 V, lower transconductance of MOS transistors, and signal attenuation introduced by the input capacitive divider. Further, it is worth noting here that input noise of the MI-OTA will be increased due to the lower bulk transconductance, being a result of an input capacitive divider combined with a BD technique. However, the input range is extended in the same proportion, and hence, the dynamic range will not be affected and will be the same as for the gate-driven counterpart of the proposed OTA [30].

### 2.2. Proposed Shadow Oscillator

Figure 3 shows the block diagram of the shadow oscillator [3]. The system consists of a three-input single-output biquad filter and three amplifiers, *A*_1_, *A*_2_, and *A*_3_. Assume that the output *V_o_* of the biquad filter can be expressed by:(6)$$V_{o}=\frac{s^{2}\varepsilon V_{1}+s\beta V_{2}+\alpha V_{3}}{as^{2}+bs+1}$$
where *ε*, *β*, *α* are real numbers of numerator and *a*, *b* are real numbers of a denominator which depends on the filter structure. The high-pass filter can be obtained if *β* = *α* = 0, the band-pass filter can be obtained if *ε* = *α* = 0, and the low-pass filter can be obtained if *ε* = *β* = 0.

Considering the amplifiers, the outputs *V*_1_, *V*_2_, *V*_3_ of the amplifiers *A*_1_, *A*_2_, *A*_3_ can be expressed respectively by: (7)$$\begin{array}{c} V_{1}=A_{1}V_{o}\\ V_{2}=A_{2}V_{o}\\ V_{3}=A_{3}V_{o} \end{array}\}$$

If voltages *V*_1_, *V*_2_, *V*_3_ are connected to the inputs of a biquad filter, the characteristic equation of the shadow oscillator can be expressed by:(8)$$\left(a-\varepsilon A_{1}\right)s^{2}+\left(b-\beta A_{2}\right)s+\left(1-\alpha A_{3}\right)=0$$

The condition and the frequency of oscillation can be expressed, respectively, by:(9)$$\frac{b-\beta A_{2}}{a-\varepsilon A_{1}}=0$$
(10)$$\omega_{o}=\sqrt{\frac{1-\alpha A_{3}}{a-\varepsilon A_{1}}}$$

Thus, the condition and the frequency of oscillation can be controlled independently by *A*_2_ and *A*_3_, respectively. The amplifier *A*_1_ can also be used to control the frequency of oscillation. However, the amplifier *A*_1_ is not significant and it can be removed for obtaining a minimum number of devices because amplifier *A*_3_ is already used to control the frequency of oscillation and it is enough to behave as a shadow oscillator. Thus, the multiple-input biquad filter that provides only low-pass and band-pass filters can be used.

Figure 4 shows the proposed shadow filters using MI-OTAs. Figure 4a shows the proposed shadow oscillator employing two single-output MI-OTAs, one dual-output MI-OTA, two grounded capacitors, and two grounded resistors. The *g_m_*_1_, *g_m_*_2_, *C*_1_, and *C*_2_ are used to realize the biquad filter. Using (1) and nodal analysis, the output *V_o_* of the biquad filter can be given by:(11)$$V_{o}=\frac{sC_{1}g_{m2}V_{2}-g_{m1}g_{m2}V_{1}}{s^{2}C_{1}C_{2}+sC_{1}g_{m2}+g_{m1}g_{m2}}$$

It should be noted that if *V*_1_ = *V_in_* and *V*_2_ = 0 (connected to ground) the low-pass filter can be obtained; if *V*_2_ = *V_in_* and *V*_1_ = 0 (connected to ground) the band-pass filter can be obtained.

The *g_m_*_3_, *R*_1_, and *R*_2_ are used to realize the amplifiers; the transfer function of the amplifiers can be expressed by:(12)$$A_{1}=\frac{V_{A1}}{V_{o}}=g_{m3}R_{1}$$
(13)$$A_{2}=\frac{V_{A2}}{V_{o}}=g_{m3}R_{2}$$

Combining (11)–(13), the characteristic equation of the oscillator can be expressed by:(14)$$s^{2}C_{1}C_{2}+sC_{1}g_{m2}\left(1-g_{m3}R_{2}\right)+g_{m1}g_{m2}\left(1+g_{m3}R_{1}\right)=0$$

The condition and the frequency of oscillation can be given respectively by:(15)$$1-g_{m3}R_{2}=0$$
(16)$$\omega_{o}=\sqrt{\frac{g_{m1}g_{m2}}{C_{1}C_{2}}\left(1+g_{m3}R_{1}\right)}$$

The condition of oscillation can be controlled by *A*_2_ through adjusting *R*_2_, and the frequency of oscillation can be controlled by *A*_1_ through adjusting *R*_1_ with constant *g_m_*_3_. Thus, the condition and the frequency of oscillation of the proposed shadow oscillator in Figure 4a can be independently controlled.

The shadow oscillator in Figure 4a is proposed for obtaining the minimum number of active elements, but the frequency of oscillation can be controlled by amplifier *A*_1_ through adjusting *R*_1_. Figure 4b shows the proposed shadow oscillator for obtaining an electronic tuning capability. It can be obtained by slightly modifying Figure 4a by adding an additional *g_m_*_4_. Thus, four single-output MI-OTAs are required. Using (1) and nodal analysis, the characteristic equation of the proposed oscillator in Figure 4b can be expressed by:(17)$$s^{2}C_{1}C_{2}+sC_{1}g_{m2}\left(1-g_{m3}R_{2}\right)+g_{m1}g_{m2}\left(1+g_{m4}R_{1}\right)=0$$

The condition and the frequency of oscillation can be given respectively by:(18)$$1-g_{m3}R_{2}=0$$
(19)$$\omega_{o}=\sqrt{\frac{g_{m1}g_{m2}}{C_{1}C_{2}}\left(1+g_{m4}R_{1}\right)}$$

The condition of oscillation can be controlled electronically by *A*_2_ through adjusting *g_m_*_3_ and the frequency of oscillation can be controlled electronically by *A*_1_ through adjusting *g_m_*_4_. Thus, the condition and the frequency of oscillation of the proposed shadow oscillator in Figure 4b can be independently and electronically controlled.

The range for tuning the frequency of oscillation in (19) in terms of 1 + *g_m_*_4_*R*_1_ may be limited because this term is in a square-rooting form. Due to the *g_m_*_4_ providing multiple-input terminals, if the additional non-inverting terminal of *g_m_*_4_ is connected to *V*_o_ (red dashed line in Figure 4b), the term 1 + *g_m_*_4_*R*_1_ of (19) will become 1 + 2*g_m_*_4_*R*_1_. The range for tuning the frequency of the oscillator in this case will be increased compared with (19).

Considering the nodes *V_o_*_1_ and *V_o_*_2_ of the proposed shadow oscillators, the relationship of *V_o_*_1_ and *V_o_*_2_ is given by: (20)$$V_{o2}=-\frac{g_{m1}}{sC_{1}}V_{o1}$$
where *g_m_*_3_*R*_1_ ≈ 1 (or *g_m_*_4_*R*_1_ ≈ 1 for Figure 4b). It can be expressed that the phase difference of signals *V_o_*_1_ and *V_o_*_2_ are 90 (π/2) degree and the magnitude is |*g_m_*_1_/*C*_1_|. It should be noted that the magnitude of *V_o_*_1_ and *V_o_*_2_ will be constant if the values of *g_m_*_1_ and *C*_1_ are not used for varying the frequency of oscillation, like in conventional quadrature oscillators [14,18,19,20].

As is well-known, sinusoidal oscillators require an amplitude stabilization mechanism, which can exploit circuit nonlinearities, limiting large-signal loop gain for larger amplitudes, or a special automatic gain control (AGC) circuit [14,18,21]. In the proposed solution, the amplitude stabilization mechanism is based on circuit nonlinearities, which provides sufficient quality for the generated waveforms, while it does not require an additional, relatively complex, circuitry to realize the AGC loop. Note that, thanks to the shadow principle, the frequency of oscillations and the condition of oscillations can be controlled independently, which makes such a simple solution possible (frequency tuning does not affect the nonlinear mechanism limiting the frequency of oscillations, resulting in their constant amplitude). Note that similar solutions, based on exploiting the circuit nonlinearities, were applied in some other shadow oscillators available in literature, such as [22,23]. However, in more demanding cases, a special AGC circuit could also be applied in the proposed oscillator. In the version described in this work, the amplitude of oscillations is approximately equal to the linear range of the used OTAs.

The effect of OTA parasitic elements on the performance of the shadow oscillator is considered using the equivalent circuit that represents a non-ideal OTA with finite parasitic resistances and capacitances in [41]. These finite parasitic resistance and capacitances will appear if the OTA operates near the cut-off frequency. As the input terminal of the proposed OTA in Figure 2 is connected to the parallel floating of high-resistance M_L_ and floating capacitance *C_B_*, the parasitic resistance and capacitance at the input terminal will be neglected. Thus, only parasitic resistance and capacitance at the output terminal *R*_o_ and *C*_o_ will be considered. Letting *C_o_*_1_//*R_o_*_1_, *C_o_*_2_//*R_o_*_2_, and *C_o_*_3_//*R_o_*_3_ are the parasitic parameters of OTA_1_, OTA_2_, and OTA_3_, respectively. Consider Figure 4a: external capacitors *C*_1_ and *C*_2_ are parallel connected at the node *V_o_*_1_ and *V_o_*_2_, thus the effects of parasitic capacitances *C_o_*_1_ and *C_o_*_2_ can be neglected if *C*_1_ >> *C_o_*_1_ and *C*_2_ >> *C_o_*_2_, while the effects of parasitic resistors *R_o_*_1_ and *R_o_*_2_ can be reduced if the operating frequency of the circuit is *ω_o_* >> max[1/(*C*_1_ + *C_o_*_1_)*R_o_*_1_, 1/(*C*_2_ + *C_o_*_2_)*R_o_*_2_)] [42]. The voltage gains *A*_1_ and *A*_2_ in (12) and (13) become *g_m_*_3_(*R*_1_//*R_o_*_3_) and *g_m_*_3_(*R*_2_//*R_o_*_3_), respectively. This parasitic resistance *R*_o3_ can be absorbed if *R*_1_ << *R_o_*_3_ and *R*_2_<<*R*_o3_ while the parasitic impedances at nodes *V_A_*_1_ and *V_A_*_2_ are (*R_o_*_3_//*C_o_*_3_), respectively. The effects of parasitic impedances at nodes *V_A_*_1_ and *V_A_*_2_ will be alleviated if the frequency of operation is smaller than 1/(*R_o_*_3_//*C_o_*_3_). However, since the circuit operates at very low frequencies, the impact of parasitic elements on the circuit operation is expected to be low.

## 3. Simulation Results

The CMOS structure of the MI-OTA and the shadow oscillator were designed and simulated in Cadence program using 0.18 µm CMOS technology from TSMC. The transistor aspect ratio is shown in Table 1. The MIM input capacitors *C_B_* = 0.5 pF. The voltage supply is 0.5 V for nominal value of I_set_ = 10 nA, and the power consumption of the two- and one-output OTA is 15.25 nW and 13.5 nW, respectively. The basic parameters of the MI-OTA can be found in [30] as follows: the DC voltage gain is 31.17 dB, common mode rejection ratio is 90.05, power supply rejection ratio is 37.26 dB and input offset is 0.224 mV with 20 pF load capacitance.

Figure 5 shows the output current and the transconductance of the MI-OTA versus the input voltage and various I_set_ = 10 nA, 20 nA and 30 nA. The high linearity with wide input voltage range of ± 230 mV is notable. For instance, for this input voltage range and for I_set_ = 10 nA (*g_m_* = 26.4 nS), the deviation of the transconductance value is below 5%. The high linearity under 0.5V supply voltage is obtained thanks to the bulk-driven multiple-input technique and the negative feedback.

The shadow oscillator was simulated with nominal I_set_ = I_set1,2,3,4_ = 10 nA, the off-chip passive elements *C*_1_ = *C*_2_ = 20 pF. The nominal power consumptions of the shadow oscillators in Figure 4 (a) and (b) were 42.25 nW and 54 nW, respectively. From (15) and (16), it can be concluded that the condition of oscillation can be controlled by *g_m_*_3_*R*_2_ (*A*_2_) and the frequency of oscillation can be given by *g_m_*_3_*R*_1_ (*A*_1_), thus *R*_2_ = 42 MΩ was selected to satisfy (15) (*g_m_*_3_ = 26.4 nS) and *R*_1_ = 100 kΩ was given the oscillating frequency in (16), whereas the core parameters of oscillator, *g_m_*_1_ = *g_m_*_2_ = 26.4 nS and *C*_1_ = *C*_2_ = 20 pF are constant. Figure 6a,b shows the running oscillation and steady state, respectively. The outputs *V_o_*_1_ and *V_o_*_2_ are in quadrature with frequency of 202 Hz. This result confirms that the condition of oscillation can be controlled by amplifier *A*_2_ (*g_m_*_3_*R*_2_).

The Fast Fourier Transform (FFT) of *V_o_*_1_ and *V_o_*_2_ is shown in Figure 7. The Total Harmonic Distortions (THDs) for *V_o_*_1_ and *V_o_*_2_ were 0.397% and 0.35%, respectively.

Figure 8 shows the relation between *V_o_*_1_ and *V_o_*_2_ that can confirm the quadrature relationship of output signals. It can be noticed that the amplitudes of *V_o_*_1_ and *V_o_*_2_ are nearly equal at frequency of 202 Hz.

Table 2 shows the impact of the temperature and process corners on the performance of the oscillator. The temperature corners were −20 °C, 27 °C and 50 °C; the process corners were fast-fast (FF), fast-slow (FS), slow-fast (SF) and slow-slow (SS). The oscillator is capable of oscillating under all temperature and process conditions thanks to the tuning capability of the condition and frequency of oscillation. The required frequency can be simply readjusted by the setting currents.

Figure 9a shows the frequency tuning capability of the oscillator in Figure 4a with I_set1,2,3_ = 10 nA versus R_1_ = 0.1 MΩ, 1 MΩ and 5 MΩ. Figure 9b shows the extra freedom of electronic tunability that was offered by the oscillator in Figure 4b with I_set1,2,3_ = 10 nA and various I_set4_ = 10 nA, 20 nA, 30 nA. This result confirms that the frequency of the oscillator can be controlled by the amplifier A_1_ (*g_m_*_3_R_1_ and *g_m_*_4_R_1_).

Note that the electronically adjustable *g_m_* via I_set_ is used in this design for fine-tuning of the oscillator frequency; hence, it enables readjusting of the required frequency in case of its deviation after fabrication. If wider tuning range is needed then the I_set_ should be increased, at the cost of increased power consumption of the application.

The amplitudes of *V*_o1_ and *V*_o2_ versus the oscillation frequencies from Figure 4a,b are shown in Figure 10a,b, respectively. It can be noticed that, thanks to the shadow principle, the amplitude of output signals is only slightly changed when the oscillation frequencies are varied by amplifiers.

Figure 11 shows the amplitude versus oscillation frequency in case of tuning I_set1,2_ = 10 nA, 20 nA and 25 nA. Here, due to the lack of the shadow principle, the decreasing amplitude with increase of the frequency is notable. Comparing Figure 11 with Figure 10, it can be concluded that the shadow oscillator that tunes the frequency by an amplifier offers less amplitude change of the output signal.

Figure 12a shows the results of Monte Carlo (MC) process and mismatch analysis, with 200 runs, for *V_o_*_1_ (a) and *V_o_*_2_ (b). The mean frequency value was 195 Hz and the standard deviation was 15.27 Hz.

The proposed shadow oscillators have been compared in Table 3 with previous works [14,21,22,23]. Both types of oscillators with amplitude stabilization based on AGC circuits [14,21], as well as circuit nonlinearities [22,23], have been selected for comparison. The proposed oscillator using an amplifier to vary the frequency has a simpler structure compared with [14,21] that use AGC circuits to control the amplitude. Compared to [22], the proposed oscillator enjoys an electronic control, and compared to [23], the proposed oscillator enjoys independent control. Compared to all previous works [14,21,22,23], the proposed structure consumes ultra-low levels of power, rendering it suitable for extremely low-voltage and low-power applications such as biomedical systems.

## 4. Conclusions

This paper presents a low-voltage and nano-power shadow oscillator based on MI-OTA for low-frequency applications. The proposed oscillator uses two-input single-output biquad filter and amplifiers. The condition and the frequency of oscillation of the shadow oscillator can be controlled electronically and independently using amplifiers. The oscillator can be tuned by resistor, capacitor and by the setting current I_set_. The simulated results show low THD (around 0.39%) for both output signals and acceptable tuning range. The 0.5 V supply voltage and the 54 nW power consumption of the oscillator is another benefit of the proposed circuit.

## Figures and Tables

**Figure 1 sensors-23-02146-f001:**
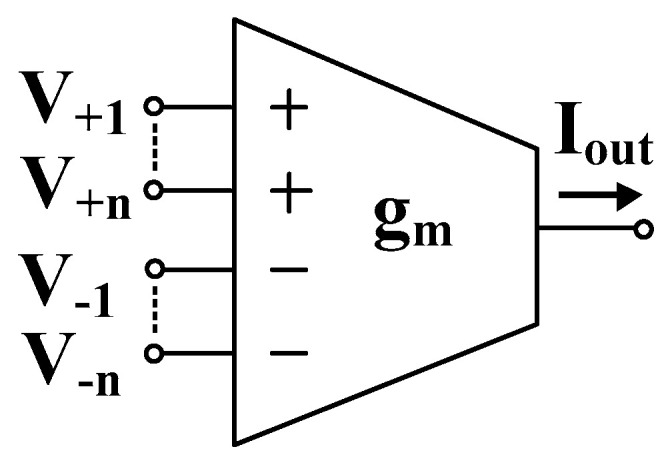
Electrical symbol of MI-OTA.

**Figure 2 sensors-23-02146-f002:**
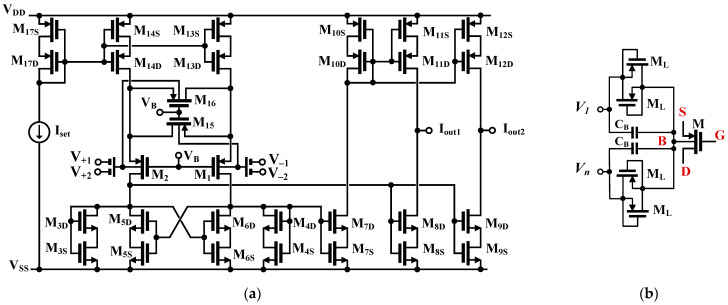
CMOS implementation of proposed MI-OTA: (**a**) schematic, and (**b**) bulk-driven MI-MOST technique.

**Figure 3 sensors-23-02146-f003:**
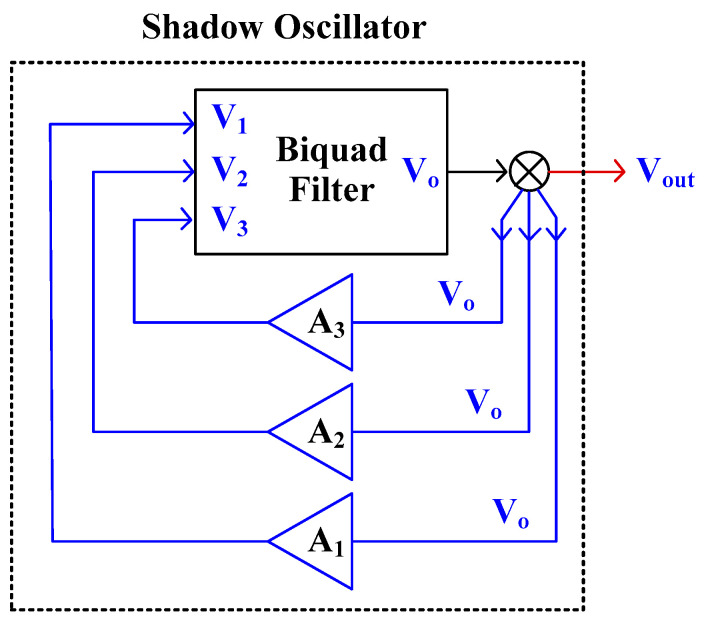
Block diagram of shadow filter.

**Figure 4 sensors-23-02146-f004:**
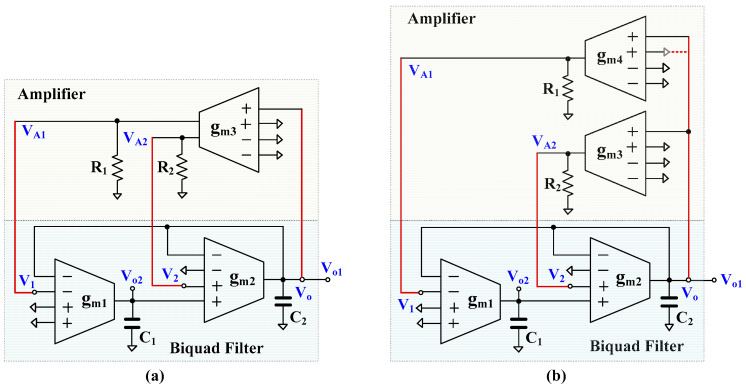
Proposed shadow oscillator using MI-OTA for (**a**) obtaining minimum number of MI-OTAs, and (**b**) obtaining electronic tuning capability.

**Figure 5 sensors-23-02146-f005:**
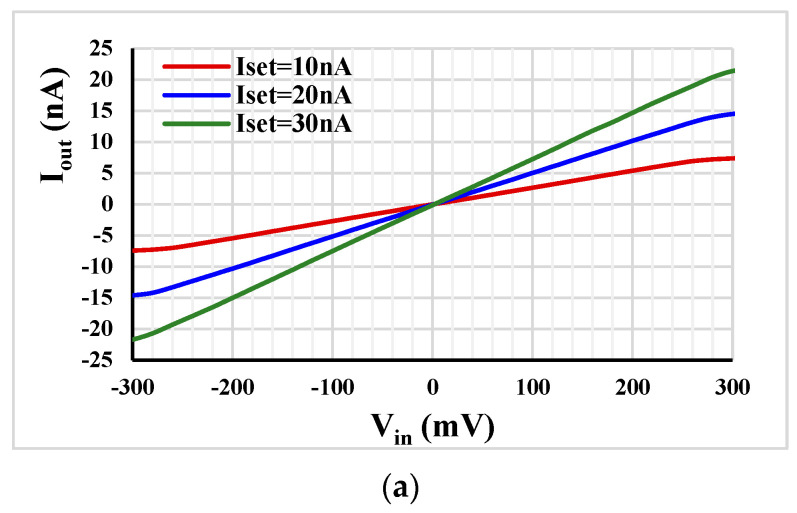
The MI-OTA output current (**a**) and the transconductance (**b**) versus the input voltage with various I_set_.

**Figure 6 sensors-23-02146-f006:**
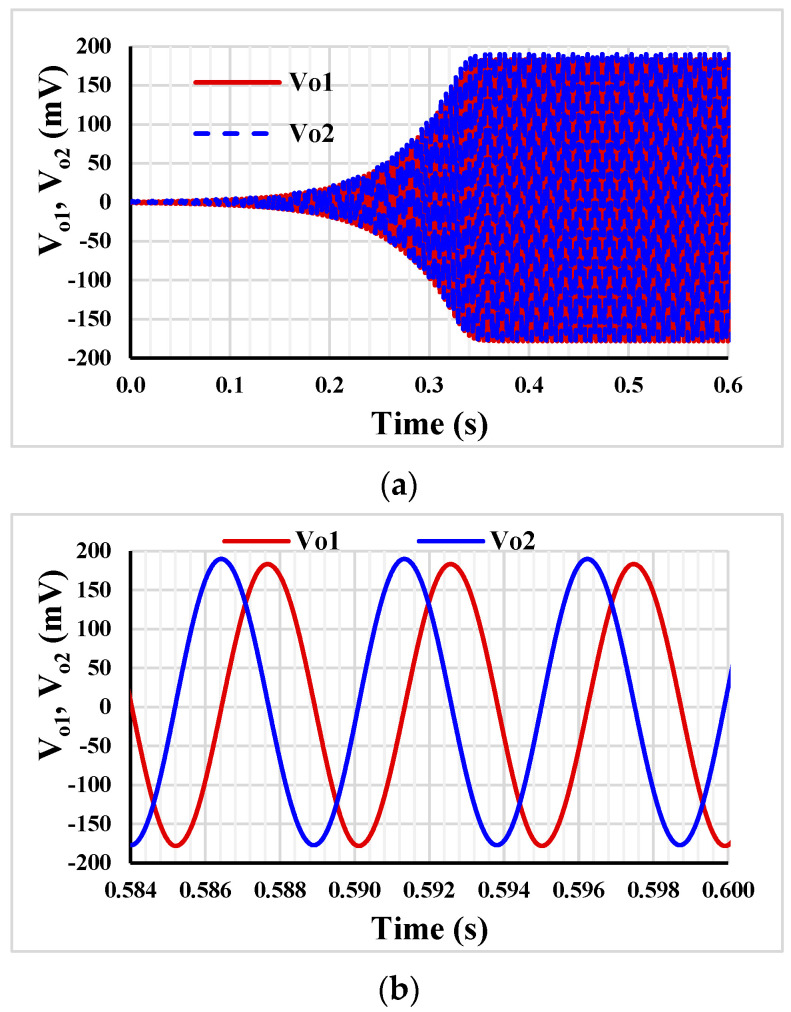
The running oscillation (**a**) and the steady state (**b**).

**Figure 7 sensors-23-02146-f007:**
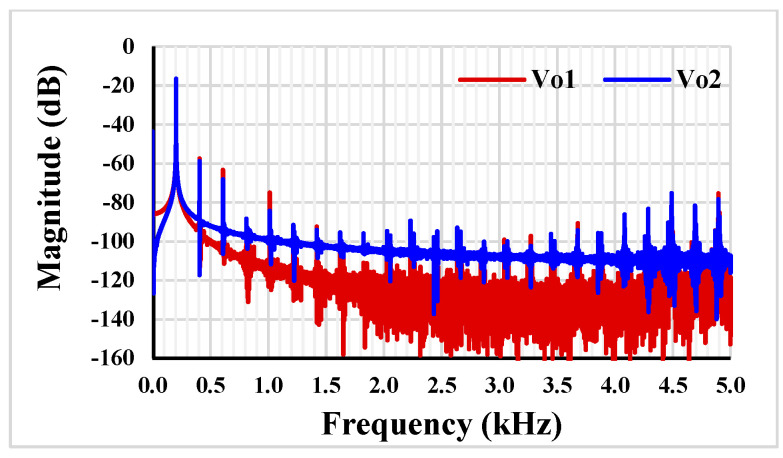
The FFT for *V_o_*_1_ and *V_o_*_2_.

**Figure 8 sensors-23-02146-f008:**
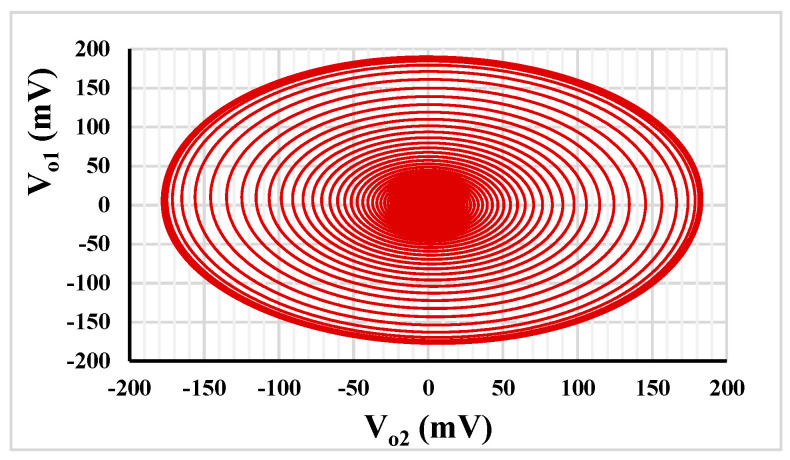
The quadrature relationship between *V_o_*_1_ and *V_o_*_2_.

**Figure 9 sensors-23-02146-f009:**
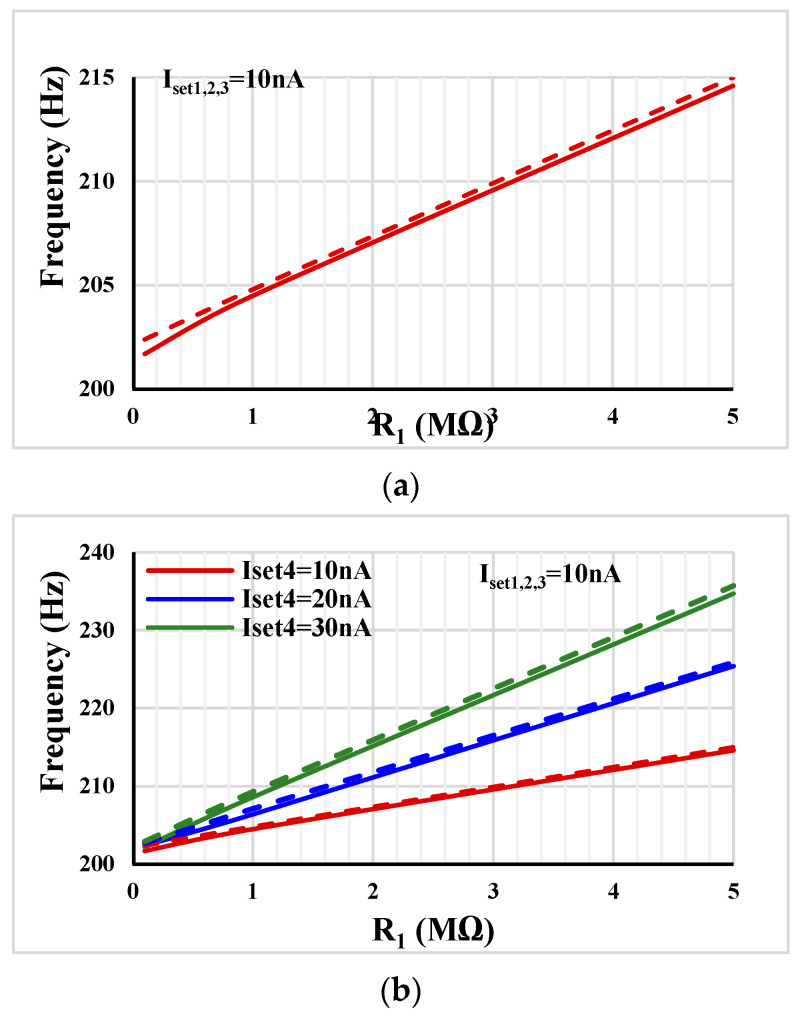
Frequency tuning capability of the shadow oscillator: (**a**) Figure 4a, (**b**) Figure 4b (*V*_o2_ dashed line).

**Figure 10 sensors-23-02146-f010:**
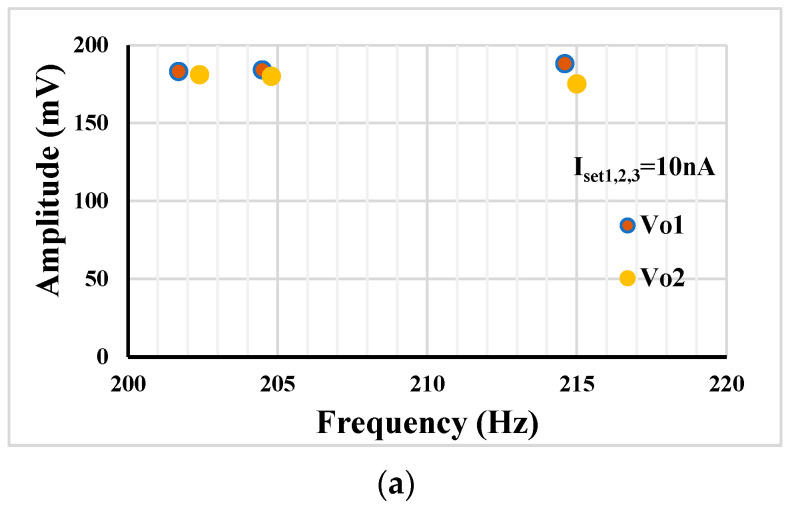
The amplitudes of *V*_o1_ and *V*_o2_ versus the oscillation frequencies (**a**) varying by R_1_, and (**b**) varying by *g_m_*_4_ via I_set4_.

**Figure 11 sensors-23-02146-f011:**
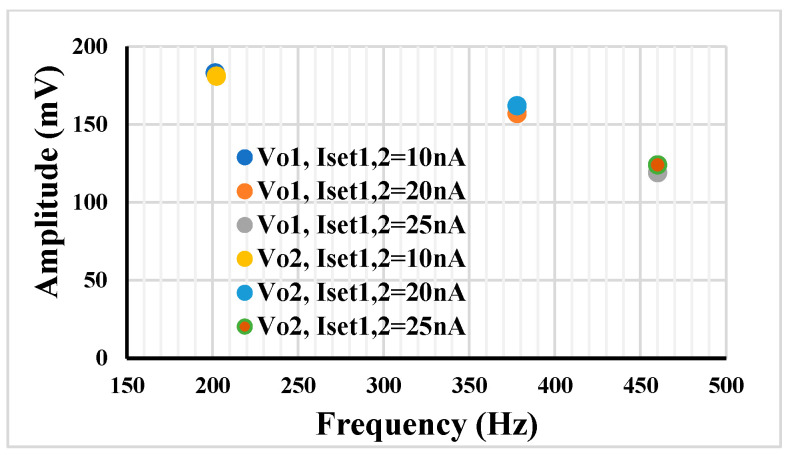
The amplitudes of *V*_o1_ and *V*_o2_ versus the oscillation frequencies when I_set1_ and I_set2_ are varied.

**Figure 12 sensors-23-02146-f012:**
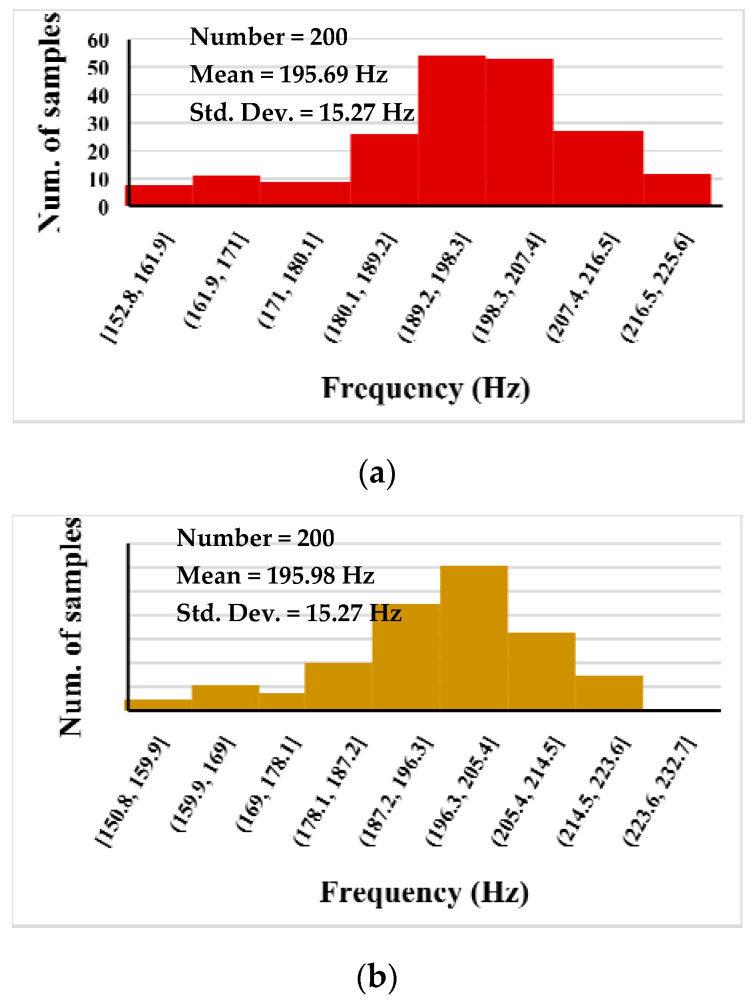
200-run MC analysis for *V*_o1_ (**a**) and *V*_o2_ (**b**).

**Table 1 sensors-23-02146-t001:** Transistor aspect ratio of the MI-OTA.

Device Name	W/L (µm/µm)
M_1_, M_2_	4 × 1.2/1.2
M_3D_, M_4D_, M_7D_, M_8D_, M_9D_	4/1.2
M_3S_, M_4S_, M_7S_, M_8S_, M_9S_	4/0.2
M_5D_, M_6D_	1.2/1.2
M_5S_, M_6S_	1.2/0.2
M_10D_, M_11D_, M_12D_	8/1.2
M_10S_, M_11S_, M_12S_	8/0.2
M_13D_, M_14D_	10/1.2
M_13S_, M_14S_	10/0.2
M_17D_	20/1.2
M_17S_	20/0.2
M_L_	5/4

**Table 2 sensors-23-02146-t002:** Temperature and process corners analysis.

	Temp.			Proccess			
	−20 °C	27 °C	50 °C	FF	FS	SF	SS
Frequency (Hz)	216	202	195	212	211	180	192
Amplitude (mV)	181	181	173	179	189	168	174

**Table 3 sensors-23-02146-t003:** Comparison of the proposed shadow oscillators with previous works.

Factor	Proposed	[14]	[21]	[22]	[23]
Realization	CMOS	Commercial IC	CMOS	Commercial IC	CMOS
No. of active elements	4-OTA	3-LT1228	1-CG-CFDOBA, 1-CG-BCVA	1-Op-Amp 2-CFOA	3-VDTA
No. of passive elements	2-R, 2-C	5-R, 2-C (Figure 5)	3-R, 2-C	7-R, 2 C	2-C
Oscillation frequency	209−235 Hz	0.08−1.1 MHz	0.25−8 MHz	1−14.25 kHz	0.265−0.323 kHz
Supply voltage	0.5 V	±5 V	±1 V	±10 V	±1 V
Power consumption	54 nW	-	-	-	-
Electronic control	yes	Yes	Yes	No	Yes
Orthogonal control of CO and FO	yes	Yes	Yes	Yes	No
Technique to control amplitude	nonlinearities	Using AGC	Using AGC	nonlinearities	nonlinearities

Note: CFOA = Current Feedback Operational Amplifier; AGC = Amplitude-Automatic Gain Control; CG-CFDOBA = Controlled Gain Current Follower Differential Output Buffered Amplifier; CG-BCVA = Controlled Gain Buffered Current and Voltage Amplifier; VDTA = Voltage Differential Transconductance Amplifier.

## Data Availability

Not applicable.

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
