# Peer review of "0.5-V Nano-Power Shadow Sinusoidal Oscillator Using Bulk-Driven Multiple-Input Operational Transconductance Amplifier"

_sensors, 2023, doi:10.3390/s23042146_

Round 1

Reviewer 1 Report

How the errors of the proposed MI-OTA influence the overall performance of the proposed circuit?

Author Response

Reviewer #1

Comments and Suggestions for Authors

How the errors of the proposed MI-OTA influence the overall performance of the proposed circuit?

Author response: Since the circuit operates at very low frequencies, the impact of parasitic elements and error of the proposed MI-OTA on the circuit operation is expected to be low. The effect of the OTA parasitic elements on the performance of the shadow oscillator is considering and already included in the paper. The text is highlighted in gray and yellow colors in the manuscript.

Reviewer 2 Report

·      The paper 0.5-V Nano-power shadow sinusoidal oscillator using bulk-2 driven multiple-input operational trans-conductance amplifier” presents a shadow oscillator.

·      Sinusoidal oscillators are frequently used in telecommunication, electronic instrumentation and control systems for producing variant generated frequencies without changing the circuit components however the amplitude get affected.

·      Shadow oscillator is used as a new technique without affecting the amplitude of the output signal, hence the shadow oscillator using multiple-input operational trans-conductance amplifiers (MI-OTAs) is proposed in this paper.

·      The CMOS implementation of the proposed MI-OTA of consisting of using two-input single-output bi-quad filter and amplifiers is as shown as it is low-frequency and low-voltage with block diagram shown in Fig. 3 that is used in the proposed shadow oscillator shown in Fig. 4 (a) obtaining minimum number of MI-OTAs and in Fig. 4(b) obtaining electronic tuning.

 Query 1.        Fig. 6 (a) is not showing Vo1 (red color) while the Vo2 (blue color) is shown, the paper be restructured as how the shadow oscillator is obtained, how it is implemented, results obtained in support of the title.

 Query 2.        The frequency range of 209-235 Hz and so the 54hW power from Table II be shown in results.

 Query 3.        The conclusions be quantified, making sure every word of the title “Nano-power shadow sinusoidal oscillator using bulk-2 driven multiple-input operational trans-conductance amplifier” has been used in the Abstract

 Query 4.        Figure 2 is used Bulk-driven against the title of Bulk-2, and how the total harmonic distortion (THD) of the output signals is calculated to be around 0.3 % 23 for 202 Hz.

 Query 5.        Most of the results of Fig. 5 to Fig. 12 are just reported, and should be discussed to support the title.

 Query 6.        Conclusion is where we say things we confirm to report things we have already reported and discussed, but its conclusion shows as if the paper not done too much.

Is the subject matter presented in a comprehensive manner?

·      The 15-page paper is presented with an enough level of flow to genuinely support the title 0.5-V Nano-power shadow sinusoidal oscillator using bulk-2 driven multiple-input operational trans-conductance amplifier”.

·      There is theoretical support and related explanation to cover the subject comprehensively, justifying the contribution in the form of results in figures from Figure 5 to Fig. 12 for the proposed architectural shadow oscillator.

Are the references provided applicable and sufficient?

·      The authors take support from forty one (41) recent journals and transaction references with none from MDPI and Science Direct.

·      The whole presentation does justify a well-deserved support to the title of 0.5-V Nano-power shadow sinusoidal oscillator using bulk-2 driven multiple-input operational trans-conductance amplifier”

Author Response

Reviewer #2

Comments and Suggestions for Authors

The paper 0.5-V Nano-power shadow sinusoidal oscillator using bulk-2 driven multiple input operational trans-conductance amplifier” presents a shadow oscillator. Sinusoidal oscillators are frequently used in telecommunication, electronic instrumentation and control systems for producing variant generated frequencies without changing the circuit components however the amplitude get affected. Shadow oscillator is used as a new technique without affecting the amplitude of the output signal, hence the shadow oscillator using multiple-input operational trans-conductance amplifiers (MI-OTAs) is proposed in this paper. The CMOS implementation of the proposed MI-OTA of consisting of using two-input single-output bi-quad filter and amplifiers is as shown as it is low-frequency and low-voltage with block diagram shown in Fig. 3 that is used in the proposed shadow oscillator shown in Fig. 4 (a) obtaining minimum number of MI-OTAs and in Fig. 4(b) obtaining electronic tuning.

Query 1. Fig. 6 (a) is not showing Vo1 (red color) while the Vo2 (blue color) is shown, the paper be restructured as how the shadow oscillator is obtained, how it is implemented, results obtained in support of the title.

Author response: Fig. 6 (a) shows both Vo1 and Vo2, however they are overlapped due to the selected time interval. The aim of this figure is to show the running oscillation. Both Vo1 and Vo2 are clear in Fig. 6(b) which is a zoom of the time interval of Fig. 6 (a).

Query 2. The frequency range of 209-235 Hz and so the 54hW power from Table II be shown in results.

Author response: Fig. 9 (a) and (b) already shows this frequency range for Iset1,2,3,=10nA, then for different Iset4=10nA, 20nA, 30nA while Iset1,2,3=10nA.

Query 3. The conclusions be quantified, making sure every word of the title “Nano-power shadow sinusoidal oscillator using bulk-2 driven multiple-input operational trans-conductance amplifier” has been used in the Abstract

Author response: The conclusion has been modified.

Query 4. Figure 2 is used Bulk-driven against the title of Bulk-2, and how the total harmonic distortion (THD) of the output signals is calculated to be around 0.3 % 23 for 202 Hz.

Author response: The title of Fig. 2 and the title of the paper use same i.e., “Bulk-driven”. The THD is calculated automatically by the professional Cadence program based on the FFT in Fig. 7.

Query 5. Most of the results of Fig. 5 to Fig. 12 are just reported and should be discussed to support the title.

Author response: More text and explanation has been added.

Query 6. Conclusion is where we say things, we confirm to report things we have already reported and discussed, but its conclusion shows as if the paper not done too much.

Author response: The conclusion has been modified.

Reviewer 3 Report

In this paper, the authors made a significant improvement in low power design of OTA, especially in low supply voltage (as low as 0.5V in 0.18um process). Although the 200 runs Monte Carlo MC analysis was made, its practicability is still doubtful. The experimental results of a chip are more persuasive to readers. Due to the extreme low supply voltage, some issues such as layout, PCB, measurement setup and so on will make a gap between simulation and test.

The manuscript was cut at line 360 with no reason. It needs to be reorganized.

Author Response

Reviewer #3

Comments and Suggestions for Authors

In this paper, the authors made a significant improvement in low power design of OTA, especially in low supply voltage (as low as 0.5V in 0.18um process). Although the 200 runs Monte Carlo MC analysis was made, its practicability is still doubtful. The experimental results of a chip are more persuasive to readers. Due to the extreme low supply voltage, some issues such as layout, PCB, measurement setup and so on will make a gap between simulation and test. The manuscript was cut at line 360 with no reason. It needs to be reorganized.

Author response: Thank you. Unfortunately, the experimental results of this circuit are not available due to limited financial budget. However, experimental results confirming the attractive futures of similar bulk-driven OTA with 0.5V was shown by us in [32].

[32] F. Khateb, T. Kulej, M. Akbari, K.-T. Tang, K.-T. “A 0.5-V multiple-input bulk-driven OTA in 0.18-μm CMOS.” IEEE Trans. Very Large Scale Integr. (VLSI) Syst. 2022, 30, 1739–1747. https://doi.org/10.1109/TVLSI.2022.3203148.

The missing text has been added.

Round 2

Reviewer 3 Report

The revised version clarifies the review and could be publsihed. 

Author Response

Comments and Suggestions for Authors:

The revised version clarifies the review and could be publsihed. 

Author answer: Thank you very much for your acceptance.